# It Is Not Easy Being Green: Recognizing Unintended Consequences of Green Stormwater Infrastructure

**Vinicius J. Taguchi [1,2,\*](ORCID), Peter T. Weiss [3], John S. Gulliver [1,2], Mira R. Klein [4], Raymond M. Hozalski [1], Lawrence A. Baker [5], Jacques C. Finlay [6], Bonnie L. Keeler [4] and John L. Nieber [5]**

1. Department of Civil, Environmental, and Geo-Engineering, University of Minnesota, Minneapolis, MN 55455, USA; gulli003@umn.edu (J.S.G.); hozal001@umn.edu (R.M.H.)
2. St. Anthony Falls Laboratory, University of Minnesota, Minneapolis, MN 55414, USA
3. Department of Civil Engineering, Valparaiso University, Valparaiso, IN 46383, USA; peter.weiss@valpo.edu
4. Humphrey School of Public Affairs, University of Minnesota, Minneapolis, MN 55455, USA; mrklein@umn.edu (M.R.K.); keel0041@umn.edu (B.L.K.)
5. Department of Bioproducts and Biosystems Engineering, University of Minnesota, St. Paul, MN 55108, USA; baker127@umn.edu (L.A.B.); nieber@umn.edu (J.L.N.)
6. Department of Ecology, Evolution, and Behavior, University of Minnesota, St. Paul, MN 55108, USA; jfinlay@umn.edu
* Correspondence: taguc006@umn.edu

**Abstract:** Green infrastructure designed to address urban drainage and water quality issues is often deployed without full knowledge of potential unintended social, ecological, and human health consequences. Though understood in their respective fields of study, these diverse impacts are seldom discussed together in a format understood by a broader audience. This paper takes a first step in addressing that gap by exploring tradeoffs associated with green infrastructure practices that manage urban stormwater including urban trees, stormwater ponds, filtration, infiltration, rain gardens, and green roofs. Each green infrastructure practice type performs best under specific conditions and when targeting specific goals, but regular inspections, maintenance, and monitoring are necessary for any green stormwater infrastructure (GSI) practice to succeed. We review how each of the above practices is intended to function and how they could malfunction in order to improve how green stormwater infrastructure is designed, constructed, monitored, and maintained. Our proposed decision-making framework, using both biophysical (biological and physical) science and social science, could lead to GSI projects that are effective, cost efficient, and just.

**Keywords:** stormwater; green infrastructure; sustainable development; ecosystem services; green gentrification; environmental justice; ponds; urban trees; bioretention

## 1. Introduction

Stormwater management is a relatively new field of study. Over the years, the stormwater management paradigm has shifted multiple times [1], and different stormwater practices have come into and out of favor. It is time to critique our work and make the effort to quantify the observed services and disservices of these systems [2].

As land development progresses and precipitation events intensify [3], the potential for urban flooding is expected to increase [4,5]. To combat stormwater-related flooding, in addition to various other associated benefits [6], many jurisdictions are encouraging the implementation of green infrastructure. Green infrastructure is the integration of ecological systems, both natural and engineered, within the built environment to maximize infrastructure, ecosystem, and community services [7]. In the

United States, the Environmental Protection Agency (US EPA) developed some of the first stormwater management recommendations and requirements through its stormwater-specific National Urban Runoff Program (NURP) from 1979 to 1983 [8]. But it was only during the early 2000s that the term "green infrastructure" began to be discussed in the context of stormwater management by the scientific community (Figure 1). Now, green infrastructure is considered an integral component of universal design principles by the American Society of Landscape Architects (ASLA) [9].

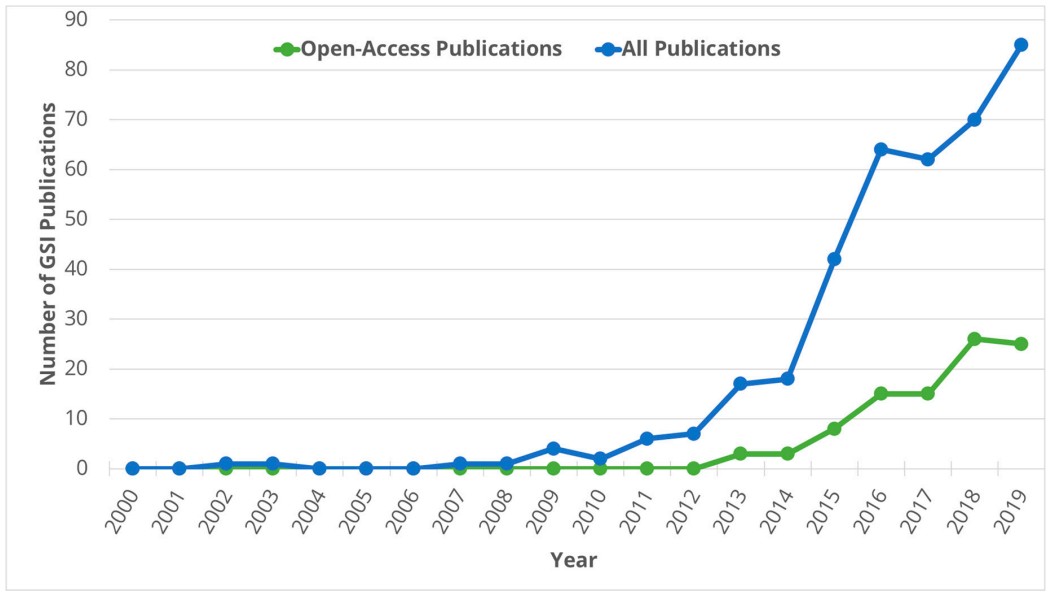

**Figure 1.** A plot of the number of scientific publications published in a given year that included either the terms "stormwater" or "storm water" within 20 words of the term "green infrastructure," based on a search of the key database Web of Science [10,11]. The blended terms "green stormwater infrastructure" and "green storm water infrastructure" were also include in the search. See Raungpan et al. [12] for a more comprehensive literature search on this topic.

Within the umbrella of green infrastructure, our focus in this manuscript is on practices used for stormwater management in urban settings, referred to here as green stormwater infrastructure (GSI), which is also sometimes called stormwater green infrastructure (SGI) or blue-green infrastructure (BGI). GSI is closely related to or encompasses the following concepts in use by different groups around the world:

- urban green infrastructure
- sustainable urban development
- low-impact development (LID)
- water sensitive urban design (WSUD)
- sustainable drainage systems (SuDS)
- sustainable urban drainage systems (SUDS)
- stormwater management
- stormwater best management practices (BMPs)
- stormwater control measures (SCMs)
- integrated management practices (IMPs)
- absorbent landscaping
- nature-based solutions

- biotreatment technologies
- urban hydrology
- green engineering
- ecological engineering
- sponge cities
- green cities
- smart cities
- green streets
- complete streets
- green corridors
- green buildings
- SmartCode
- Living Building Challenge
- Leadership in Energy and Environmental Design (LEED)

GSI provides many benefits [7,13,14] and is sometimes viewed as a cost-effective supplement to the repair, replacement, and upgrading of aging and degrading drainage infrastructure [15], which can be cost-prohibitive [16]. In the Fourth National Climate Assessment, costs for public infrastructure projects to improve wastewater conveyance (sanitary sewers) and treatment facilities, correct combined sewer overflows, and address water quality-based stormwater management in the United States were estimated at USD271 billion over a 20-year period [3]. Due to the growing interest in GSI, it is easy to overlook potential trade-offs and unintended consequences that may accompany infrastructural attempts to address flooding and urban water quality issues. Yet it is crucial to acknowledge that even natural amenities are not neutral and provide both services and disservices; as Gould and Lewis stated, low-impact development is not no-impact development [17].

There is extensive historical precedent demonstrating negative consequences resulting from celebrated infrastructure projects, even those with altruistic intentions [18,19]. During the 19th century in the United States—following the adoption of flush toilets—the decision to develop combined sewers (sewage and stormwater in the same pipes) rather than separate sewers in most cities led to widespread epidemics of typhoid and cholera in downstream cities [20], the burden of which was disproportionately felt by communities of color [21,22]. Modern wastewater treatment plants servicing combined sewer systems can be overwhelmed during large storm events and discharge untreated sewage into receiving water bodies. Combined sewer systems were not designed with future wastewater treatment concerns in mind and remain in use in many cities because separating sewer networks is often cost-prohibitive [23]; much contemporary stormwater management in such areas now centers around preventing combined sewer overflows. New GSI developments should be approached with historical lessons in mind by considering the potential for unintended consequences.

Similarly, there is an emerging body of research on the shortcomings and limitations of GSI [24–27], and we now know that "green" is not the same as "infallible." For example, rain gardens have become a common, if not popular, GSI methodology used in many stormwater management plans. While rain gardens are often successful in reducing metal concentrations in stormwater, a simultaneous net export of phosphorus from rain garden sediments [28–30] has been observed. Other studies have shown that the long-term stormwater treatment capability of wetlands is questionable. Data suggests that, under certain conditions such as low dissolved oxygen, wetlands can be a source of nutrients to downstream waters [31,32]. Successful implementation requires the consideration of diverse factors, both scientific and social in nature.

While sustainable development has incorporated both the language of GSI and social equity in its rise to prominence, the two have yet to be adequately considered in tandem [33,34]. Doing so requires an analysis of both the procedural (who participates in decision making) and distributional (who benefits, and how) aspects of environmental justice. Decision-making processes inevitably exist in a context with unequitable distribution of the power to select decision makers, what concerns are deemed relevant, and who benefits from the outcome [35]. Socially vulnerable communities (lower income and non-White populations) are often located in higher-risk areas prone to flooding and other hazards [19]. Disinvestment in flood-protection infrastructure for these areas often resulted from cost-benefit analyses that favor protecting higher-income (and more likely White) communities [36]. Now, the same flood-prone areas are being targeted for GSI development that prioritizes temporary flood storage, results in large population displacement, and provides few amenities to those affected [37]. New GSI affects an existing built environment and socioeconomic context, and a conscious effort must be made if GSI benefits are to be distributed equitably [38].

The increasing use of collaborative (working "with" the community) and bottom-up (working "for" the community) decision-making processes is another approach toward distributing GSI benefits to all and preventing displacement [33,39,40]. The inequitable distribution of GSI in the context of historical and ongoing disinvestments in socially vulnerable communities have contributed to disparities in health and other measures of well-being. Environmental justice also necessitates a consideration of green gentrification, wherein GSI implementation can lead to rising property values and rents, displacement

of historical communities in the name of flood protection, and a loss of a sense of community belonging or identity [38,41]. The benefits of GSI are mediated by complex social, ecological, and technical factors, which can determine where and when nature-based solutions deliver net benefits or costs to different social groups [13,42]. Therefore, GSI planning and design decisions must consider environmental justice issues.

A Framework for GSI Decision Making

We introduce a new framework for planners implementing GSI that integrates both biophysical (biological and physical) science and social science (Figure 2) to increase the probability of outcomes that are effective, cost efficient, and just. Our framework provides a guide for prioritizing GSI project goals while remaining considerate of the broader context around the project and how different social groups may be impacted. Success depends on strong community partnerships playing an active role in each step of the planning process. This reflective and iterative approach to GSI project implementation attempts to consider project impacts and long-term resiliency early in the planning process so as to maximize the likelihood of success and minimize the likelihood of unintended consequences.

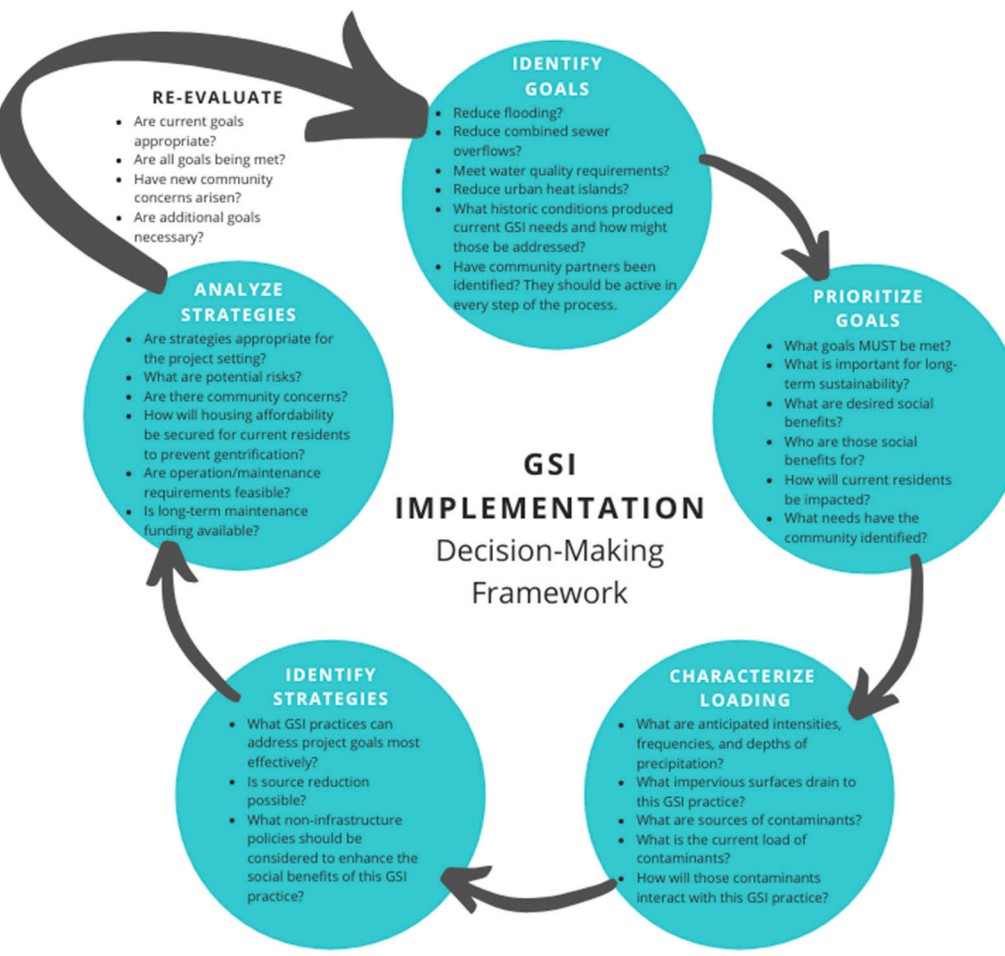

**Figure 2.** Decision-making framework for equitably and effectively implementing green stormwater infrastructure (GSI).

The discussion below is focused on the most widely implemented GSI practices with the greatest potential for unintended consequences, keeping in mind that our discussion is limited to our experiences (primarily in cities in temperate regions of the United States) but are likely applicable in other regions around the world [43,44]. The next section on *Urban Stormwater* introduces the primary contaminants of concern in stormwater runoff. The *Past, Present, and Future of Green Infrastructure* section discusses general concepts applying to all GSI practices. And the remaining sections focus on specific GSI

practice types: *Urban Trees*, *Stormwater Ponds*, *Filtration Practices*, *Infiltration Practices*, and *Rain Gardens and Green Roofs*. Each section is prefaced by an overview exploring a system understanding of how each GSI practice fits into the hydrology and pollutant dynamics of a greater watershed. It is our hope that continuing our conversation on GSI within this framework will help GSI be implemented effectively and without inadvertently creating new and more complex challenges.

## 2. Urban Stormwater

*Overview*

Water initially enters a watershed as precipitation, primarily in the form of rain, snow, or sleet (Figure 3). Although it may acquire some contaminants from the atmosphere during its descent, this liquid or frozen precipitation enters the watershed in a relatively pollutant-free form. As this rainwater or meltwater moves across an urban landscape as stormwater, however, it picks up a variety of different pollutants. Common pollutants include nutrients, metals, suspended solids (or total suspended sediments, TSS), chloride/salinity (or total dissolved solids, TDS), polychlorinated biphenyls (PCBs), pathogens, pesticides, and heat [45]. These are subjected to regulatory threshold concentrations mandated by Phases I and II of the 1987 Clean Water Act (in the United States). With a leading cause of surface water pollution (in developed countries) being stormwater runoff, the need to improve stormwater quality is vital. This section discusses common stormwater pollutants including their sources and negative environmental impacts.

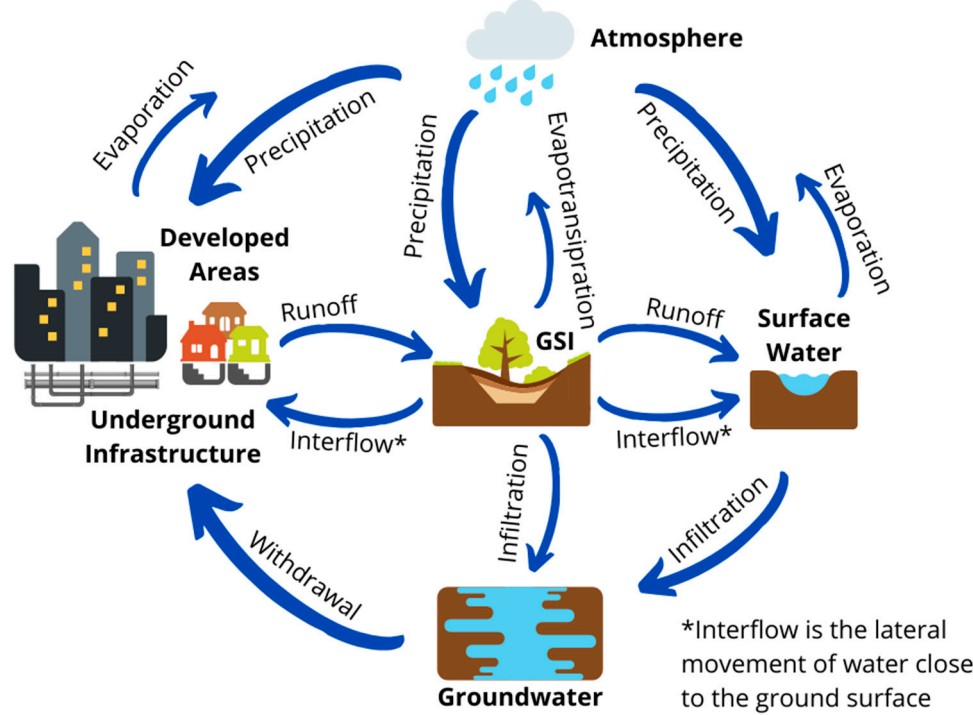

**Figure 3.** Conceptual diagram of water flow paths in the context of green stormwater infrastructure (GSI). Figure includes licensed resources [46,47].

Urban stormwater pollutants can originate from vehicles, roads, buildings, lawns, domesticated and wild animals, industrial parks, and other human-made structures [48–51]. Vehicles can contribute solid metal particles through tire and brake pad wear and the rusting of parts. They can also contribute oils and greases [52]. The roofing and siding materials of buildings can leach metals to stormwater [51]. Lawns can contribute solids through grass clippings and other vegetative matter, and, when these decompose, they can release nutrients [53]. Fertilizer from urban turf areas also adds nutrients to runoff. In addition, road salt contributes considerable chloride to urban waters [54]. Pet feces that

is not properly managed can provide an additional source of nutrients and pathogens to runoff [49]. Finally, industrial parks contribute pollutants that correspond to the industrial processes that occur on-site and can include metals and organic chemicals [55–57].

Various studies have attempted to characterize typical pollutant concentrations in urban stormwater relative to drainage area land-use and various other factors [58–65]. Although GSI practices receive many of these pollutants, specific practices may be specifically suited to treating different pollutants and each practice has its own performance limitations [66,67]. Therefore, GSI design, implementation, and maintenance must target specific pollutants of interest if the practices are to be effective [68]. The following section on the *Past, Present, and Future of Green Infrastructure* discusses general concepts applying to all GSI practices.

## 3. Past, Present, and Future of Green Infrastructure

### 3.1. Overview

If rain was to land on a natural, vegetated surface, it could likely infiltrate into the soil, be taken up by plants, or run off (especially during large rain events). Rain landing on a paved surface is more likely to run off and become urban stormwater, which, if left untreated, carries contaminants directly into rivers, lakes, oceans, or other water bodies. The goal of GSI is to allow the journey of stormwater to more closely approximate pre-development conditions, such as by allowing opportunities for plant uptake and infiltration [69]. Each GSI practice goes about this in different ways.

### 3.2. Gray Green Infrastructure Spectrum

GSI exists on a spectrum between natural ecosystems providing services and "gray" stormwater control measures that rely on conventional strategies of drainage, retention, and detention without relying on living organisms (Figure 4). A thorough discussion of the key differences between GSI and "gray" infrastructure is presented by Boyle et al. [7]. GSI strategies include evaporation, transpiration, biological absorption, storage, settling, filtration, infiltration, chemical adsorption, and reuse. These treatment strategies are affected by external factors that influence treatment rates. Settling and storage depend largely on gravity, for example, while evaporation and transpiration depend on solar energy, ambient temperature, and wind exposure. The overlapping use of diverse treatment strategies and the external and internal factors affecting them mean that each type of GSI practice—the International Stormwater BMP Database [70] currently lists 18 categories of practice types—has its strengths and weaknesses. There is a tendency to search for one-size-fits-all solutions, but trying to solve diverse problems with a single method can result in not fully addressing any of the problems.

Varied performance expectations and conflicting goals for GSI can lead to the perception of a failed implementation [13]. For many GSI projects, assumed performance values (for example, the frequently-used software EPA SWMM (US EPA Storm Water Management Model) uses a 50% phosphorus reduction value representing the median performance of stormwater ponds [71]) are applied in all scenarios and left unchanged throughout the life of the GSI practice. A list of various English-language stormwater guidance documents from different countries is included in Erickson et al. [72]. Considering best-case and worst-case performance values could better account for variable site conditions and changes over time (for example, the Minnesota Stormwater Manual lists 34%, 50%, and 73% total phosphorus removal rates for low-, medium-, and high-performing stormwater ponds, respectively [73], and the Massachusetts Stormwater Handbook lists a 30%–70% range [74]). Assuming a range of realistic performance values does add some complexity to demonstrating progress toward regulatory objectives, but considering them at the planning stage also increases the likelihood that treatment goals will be met and continue to be met over the project lifespan [75].

Stormwater management requirements vary by jurisdiction and in some areas are altogether absent. Even where regulations exist, on-site stormwater management may not be required until the area of land under development by a single project exceeds a certain threshold (varies by jurisdiction) and

may not come into effect for single-parcel redevelopments. In watersheds where GSI is lacking, it is not uncommon for the existing GSI to receive drainage from areas drastically in excess of what the GSI can effectively treat. While undersized practices still offer some treatment benefits, such design limitations may impact longevity, maintenance requirements, and the resilience to accommodate extreme rainfall events. Shortfalls in design, maintenance, and resilience of GSI can have extensive material impacts on nearby communities. Racialized historical disparities in gray infrastructure planning have disproportionately exposed minority and low-income residents to flooding, combined-sewer overflows, drinking water contamination, and other public health risks [76,77]. Yet, contemporary implementation of GSI practices risks reproducing these disparities in infrastructure access, particularly given the lack of socioeconomic consideration in existing GSI modeling [78]. Research from Portland (Oregon, USA) and Baltimore (Maryland, USA) offers strategies for distributing GSI features in relationship to race and income, among other factors [79].

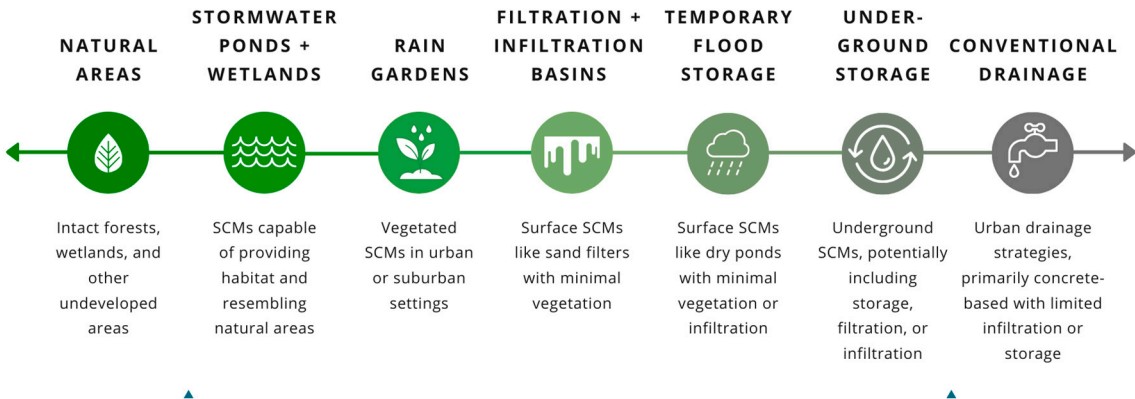

**Figure 4.** Spectrum of stormwater infrastructure from "green" to "gray." Adapted from McPhillips and Matsler [8].

### 3.3. Future Challenges

Climate change is expected to increase the frequency and intensity of extreme weather events [3] and historical measured increases in daily extreme precipitation have exceeded model projections [80,81]. While many regions are facing increased and intensifying periods of drought [82–85], the current trends in many other parts of the world are for increases in both the frequencies and magnitudes of not only extreme storm events but storms of all sizes [86]. Loading of urban stormwater contaminants, such as phosphorus, are expected to increase in colder climates because of increased winter rains [3,87]. Although loading is expected to decrease in arid climates from decreased rainfall and increased evaporation, concentrations in stormwater will be higher and continue to pose a risk [88].

Two gray infrastructure approaches to dealing with an increase in large storms are to enlarge storm sewers and build underground storage, but each of these solutions is relatively expensive. GSI strategies occasionally include some underground storage components, but using a diverse array of stormwater management tools can reduce the size and costs of drainage and storage infrastructure [15,89].

Extreme rain events can cause catastrophic damage to green and gray urban infrastructure, especially when the infrastructure is undersized or has deteriorated with age. This issue is aggravated by infrastructure designed for the watershed conditions at the time of construction rather than factoring in future development [19] and future weather patterns [5,90]. Fortunately, GSI practices have additional resilience and adaptation potential and can help alleviate undersized stormwater infrastructure, particularly when the practices can also reduce water volume such as infiltration basins,

swales, and bioretention practices. GSI also has the unique ability of undoing much of the ecological disruptions inflicted by conventional drainage and flood prevention. Highly impacted urban streams, for example, can see improved stability and water quality with GSI implementation [91,92].

## 4. Urban Trees

### 4.1. Overview

Perhaps the most intuitive way to "naturalize" or "green" an urban watershed is to plant trees. Urban tree cover is currently greatest where population densities are low [93]. Nowak and Greenfield [94] found that urban tree canopy cover (the area covered by tree leaves) in the United States decreased significantly in 23 states/districts (and did not increase significantly in any states/districts) between 2009 and 2014. This decrease was attributed primarily to aging—and, likely, unhealthy [95–98]—trees not being replaced and increasing land development. Despite reductions in tree canopy, urban trees continue to hold high social and economic value [99].

Urban trees provide stormwater benefits by intercepting rainfall and taking up water soaking into the soil around their roots [100], which can stabilize soil and reduce erosion. Trees also produce organic matter, however, in the form of leaves, twigs, pollen, seeds, fruits, and nuts, and some forms are nutrient rich. Runoff encountering urban tree litter transports much of this organic matter and associated nutrients to storm drains [62,87], although this may not represent a major fraction of the nutrient load in all watersheds [64]. Thus, for the stormwater benefits of trees to be fully realized, the organic matter and nutrients from falling leaves and other debris must also be considered in tree placement strategies and street sweeping plans.

### 4.2. Intended Purpose

Urban trees have strong social and health benefits [101,102]. For example, Sullivan et al. [103] showed that the presence of trees and grass in inner urban areas increases the amount of social activity, compared with non-vegetated areas. Trees are widely cited as providing stormwater benefits by intercepting rainfall and promoting evapotranspiration (the process of water transferring from the land to the atmosphere via evaporation and plant transpiration) [104], but these benefits are typically a secondary priority to other desired environmental benefits (such as providing species habitat) and social benefits (such as providing recreational green spaces) [44]. Trees also provide substantial cooling of urban areas [99,105], using trees and vegetation to partially mitigate heat island effects.

### 4.3. Potential Negative Consequences

Urban trees, an important site of aesthetic, social, mental health, and financial value, are often explicitly named in the literature on green gentrification [106]. There is a strong spatial correlation between tree canopy and median household income [107], for example, because urban trees are treated as an urban environmental commodity [108]. In other words, planting trees can increase property values [109], which becomes a challenge for lower-income residents, particularly in rented housing where no rent-stabilization unless affordable housing programs are established prior to tree plantings [110]. Residents may also be resistant to tree-planting efforts because of past experiences dealing with poorly maintained or neglected urban forests. Resident surveys conducted in Detroit (MI, USA) found that low-income neighborhoods had negative perceptions of tree-planting campaigns, in part because communities had to shoulder the maintenance costs of Detroit's past tree-planting efforts when budget cuts reduced the maintenance budget of the program [111].

Trees that border streets contribute substantial organic matter and nutrients—both as bulk solids and as particles including leaves, pollen, flowers, nuts, and fruits—to streets during periods of litterfall, especially in spring and fall. In a study involving 400 sweeping events over two years, Baker et al. [112] showed that removal of nitrogen and phosphorus by sweeping varied in direct relation with the amount of tree canopy overhanging streets and that enhanced sweeping (more than needed for routine

maintenance) could be highly cost-effective for removing phosphorus when targeted in time and space (for example, costing on the order of USD200 per kilogram of phosphorus removed for a street with 19% tree canopy cover). Since then, several studies have shown relationships between concentrations of nitrogen and phosphorus and tree canopy cover directly over impervious surfaces [62]. Selbig [113] showed, in a paired watershed study (one swept weekly and one swept twice during the season), that sweeping greatly reduced nutrient loading to stormwater. Janke et al. [62] demonstrated strong relationships between tree canopy cover over streets and runoff concentrations of phosphorus and nitrogen. Though less well studied, winter leaching from tree leaves may also be a major source of nutrients to winter stormwater export [87]. Coarse organic solids from trees are likely a major source of solids to storm drains, GSI, and surface waters, increasing maintenance costs [114].

Many trees produce pollen, which causes allergies and may trigger asthma attacks, and affects 8% of the United States population—pollen disproportionately affects Black and Puerto Rican residents [115]. High rates of evapotranspiration by trees combined with low groundwater recharge in urban environments can lead to land subsidence (settling), which can damage structures [116,117]. And tree roots have also been known to enter storm and sanitary sewers [98,118]. Other potential negative consequences of trees include: bird droppings (as a result of improved wildlife habitat), sap dripping on motor vehicles, and increased sidewalk damage and associated maintenance costs from roots [119,120]. These influences have not been well studied, and neither have the region-specific behavior of urban trees with respect to ecosystem services. Although urban trees are often used to mitigate urban heat island effects, their ability to provide city scale cooling may be limited in wet, tropical climates where high humidity reduces the cooling effect of evapotranspiration [121]. Shading still provides localized cooling and countless other benefits to pedestrians, but it is important to keep context/setting in mind and realize that effective tools to solve a specific problem in one region may not be as effective in another region.

*4.4. Recommendations*

The complex interactions between trees and their environments, as well as the varied reasons for which trees are planted, mean that expert knowledge on urban trees is scattered across several fields of study. These fields need to be brought together to understand these diverse benefits and negative consequences of urban trees in the context of green stormwater infrastructure.

The increasing emphasis on enhancing tree cover in cities combined with the emergence of research demonstrating negative effects of trees and urban vegetation on water quality in cities indicates that adjustments to implementation and management are necessary. Targeted, enhanced sweeping was found to be a cost-effective BMP for streets [112] in Prior Lake, Minnesota (USA), leading to a new study to develop a crediting process (in progress) for US EPA Total Maximum Daily Load (TMDL) requirements. It seems reasonable that enhanced street sweeping could be beneficial in other regions, but further study is needed to tailor the practice to local conditions. Finally, further study is needed to understand the role of coarse organic matter in stormwater maintenance.

Social science research can illuminate reasons why residents may be resistant to urban tree campaigns, ranging from concerns with maintenance costs to crime to pollen or allergen concerns [26,111]. Community engagement efforts that allow for residents to identify their values and preferences for urban trees can promote buy-in for GSI practices, especially when they are coupled with maintenance, programming, and financial support that address community concerns and ensure that benefits from urban trees flow to the residents that can most benefit from urban ecosystem services [111].

## 5. Stormwater Ponds

*5.1. Overview*

As stormwater runoff continues to flow through an urban watershed and accumulate pollutants and volume, it is likely to encounter one of the oldest and most common stormwater management

practices: the stormwater pond. While many ponds were constructed for the sole purpose of stormwater management [122,123], many were retrofitted from natural ponds and wetlands by connecting them to storm sewers and adding outlet control structures to increase water storage. Ponds are still one of the most common stormwater management practices in many regions [123–128] and yet, despite extensive study, the processes that regulate the fate of pollutants in stormwater ponds are not well understood.

Ponds slow and temporarily store stormwater runoff, allowing suspended solids containing the majority of stormwater contaminants to settle to the bottom of the pond from which they must be removed during infrequent maintenance [72]. If ponds are neglected, however, which is often the case [126,129], sediments, organic matter, and contaminants accumulate [130–132]. Under certain conditions, these contaminants may dissolve into the water and flow out of the pond [133,134]. It is therefore possible that the concentrations of contaminants are higher in the pond water than in the stormwater entering the pond [130,135]. Proper design and maintenance will avoid creating the conditions that can release accumulated contaminants. And intercepting contaminant-rich suspended solids and organic matter before they enter the pond using maintenance-friendly pretreatment devices will reduce the accumulation rate of contaminants.

### 5.2. Intended Purpose

Stormwater ponds provide temporary storage to alleviate flooding [124]. During this time, suspended solids and particle-associated pollutants, including phosphorus, can settle to the bottom of the ponds. Performance expectations are based on the incoming pollutant load, permanent pool volume, and retention of sediment [136]. Often, stormwater ponds are also expected to remove nitrogen through sedimentation and denitrification [75,137–139]. In addition to removing pollutants, stormwater ponds also provide ecosystem services (such as carbon sequestration and biodiversity) and cultural services (such as recreation and education, when properly designed and located) [14,140]. Stormwater ponds are considered by many to be largely passive systems, where maintenance is necessary only once sediment accumulation has become excessive, and typically are not regularly maintained [129]. As such, many stormwater ponds have not been designed or constructed to simplify maintenance. Furthermore, operation and maintenance practices are often specific to municipalities, states, or regions and thus vary widely in frequency, strategy, and effectiveness [72].

### 5.3. Potential Negative Consequences

Municipalities and various agencies tasked with managing stormwater ponds often face competing management priorities and performance goals. Many ponds are not maintained as treatment practices, but rather allowed to become overgrown and develop a "natural" appearance that can change the management expectations of adjacent landowners [72]. Overgrown vegetation surrounding ponds impedes access for maintenance and shelters the water surface from wind mixing and oxygenation [141, 142]. Stormwater ponds have varied suitability as ecological habitats [138,140,143,144] and may struggle to serve as both a valued water feature and a pollutant-capturer [124,145].

Stormwater ponds can have negative impacts on property values and raise safety concerns for local residents. One study in a suburb of Baltimore, Maryland found that homes adjacent to stormwater ponds had sale prices 13%–14% (approximately USD28,000) lower on average than homes not adjacent to stormwater ponds [146]. Many stormwater ponds only receive attention and maintenance when they become a nuisance to adjacent homeowners [72] due to odors, duckweed cover, or harmful algal blooms (HABs) that can even kill pets [147]. Stormwater ponds and other GSI practices can also become breeding grounds for mosquitoes [129,147–150]. Yet, pond inspections are often minimal and infrequent [72].

When ponds are maintained, the conventional approach is sediment dredging. The accumulation of organic pollutants (such as polycyclic aromatic hydrocarbons, PAHs [136,151]), heavy metals, and other toxins, however, can cause dredged sediments to be regarded as hazardous waste [72]. This results in higher transport and disposal costs, often making pond maintenance prohibitively

expensive. And yet, unmaintained ponds are not likely to function as designed and may not only fail to capture incoming particles and associated pollutants but also release previously-captured pollutants, risking contamination of receiving water bodies.

Even unmaintained ponds may continue capturing pollutants for many years until water quality begins to degrade. Continued land development may increase runoff loading to stormwater ponds beyond design expectations, causing them to accumulate sediment faster than anticipated [124]. While sedimentation rates and available volume alone typically define the design lifespan calculation, the ability of stormwater ponds to continue meeting treatment targets over their entire design lifespans is unknown. Yet, flood prevention ability often overshadows sediment removal capacity as a performance indicator [129].

Phosphorus accumulation in sediments can be released from the pond sediments into the water under conditions of high phosphorus loading and low oxygen. This "internal loading" can cause the effluent from a stormwater pond to be higher than the inflow [135,146]. Internal loading in ponds can be triggered by low dissolved oxygen concentrations, which may be worsened by the lack of mixing due to wind sheltering from tall vegetation around small ponds [141,142]. Ensuring conditions that promote phosphorus capture rather than phosphorus release is a major challenge for stormwater pond managers.

*5.4. Recommendations*

In moving toward re-thinking stormwater ponds as GSI, design practices should explicitly identify both pollutant removal goals and desired ecological benefits. Ponds designed to, for example, provide shoreline bird habitat, recreational opportunities, fish habitat, etc. must maintain good water quality. On the other hand, stormwater treatment ponds will receive untreated stormwater rich in contaminants to protect downstream water bodies of greater ecological, economic, and sentimental value. With respect to reducing pollutants downstream, and reducing maintenance costs, watershed managers should look upstream toward reducing sources of pollutants, such as chloride and suspended solids, that impair the pollutant removal function of GSI; one strategy would be to reduce the catchment area to pond area ratio, which is often ten to a hundred or more [138]. Once priorities are established, specific steps can be taken to meet performance goals.

Regular inspections, maintenance, and monitoring are necessary for any GSI practice. For example, tall vegetation could be discouraged in order to promote wind mixing and natural water aeration for phosphorus control [72]. Where trees have become established, it may be possible to enhance wind mixing by strategically removing swathes of trees on opposite shores to create corridors for prevailing winds to pass (Figure 5). This increased wind exposure could yield the additional benefits of reduced mosquito activity (such as blood feeding, breeding, and egg-laying) [147,152]. Public perception issues may still arise when removing mature trees surrounding older ponds due to the perceived destruction of "natural" ecosystems [129], but these could be tempered by planting the wind corridors with new, more diverse habitats based on low-lying, native pollinator species (Figure 5), similar to the addition of prairie strips in agricultural operations [153]. Favoring pollinator gardens over turfgrass may also discourage geese and other waterfowl from contributing nutrients to the pond [154,155]. And the visual impact of tree removal can be greatly reduced by conducting these operations in the winter when leaves are absent (from deciduous trees). Still, reduced wind sheltering alone may not be enough to improve water quality in ponds with established histories of sediment phosphorus release [134,156,157]. Such large-scale maintenance is also costly and less preferable than regular, preventative maintenance [72].

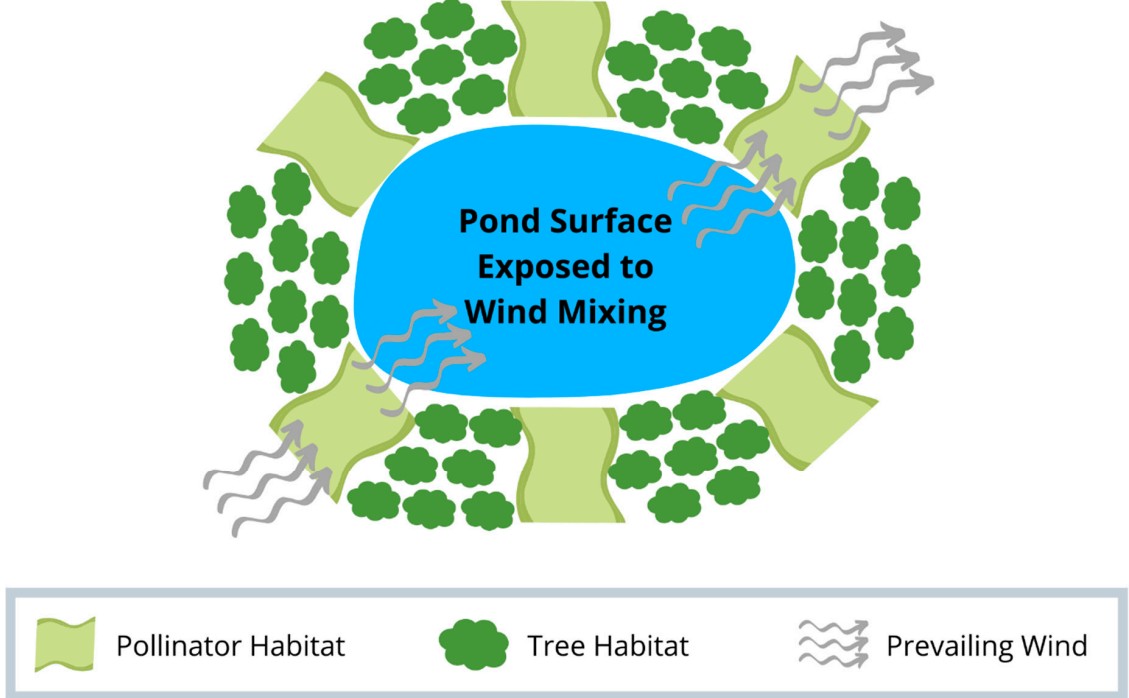

**Figure 5.** Diagram of tree removal scheme to reduce wind sheltering in older ponds where mature trees are removed and replaced with low-lying, native pollinator species that allow prevailing winds to pass uninterrupted across the surface of the pond [158].

One way to support regular maintenance is to make it simple and inexpensive. Pretreatment practices, such as street sweeping, forebays, swales, bioswales, baffles, hydrodynamic separators, and more, can capture a large fraction of the suspended solids and leaves containing a portion of the overall contaminant load and be routinely emptied using a vacuum truck. Such pretreatment can be relatively inexpensive and can prevent or delay the need for costly dredging, particularly in newer or recently-dredged ponds where contaminant-rich sediments have not already accumulated. Underground pretreatment practices can keep trash and debris from stormwater [159] out of ponds, but it is difficult to notice when such practices need to be maintained unless they are inspected regularly.

Once contaminant concentrations have accumulated in a pond, pretreatment alone cannot prevent internal loading [160]. In-pond treatments, such as aeration or oxygenation and chemical treatments of aluminum [161], iron [162], lanthanum [163], and other elements, can effectively prevent or stop internal loading when properly applied, but may be cost-prohibitive for lower-priority ponds.

Stormwater ponds appear to best manage phosphorus and other pollutants when not subjected to excessive loading rates. For this reason, low-maintenance stormwater ponds may be possible in upstream portions of watersheds draining small surface areas. Stormwater ponds are often designed to receive stormwater runoff from large drainage areas [138], however, leading to rapid sediment and pollutant accumulation. Any ponds located further downstream in a watershed and receiving runoff from large areas would require much more intensive maintenance and pretreatment. A modeling study by Emerson et al. [164] suggests that the ability of stormwater ponds to reduce peak stormwater flows during large storm events (100-year storm with a 24-hour duration; 21-centimeters of precipitation) at the watershed scale may also be limited unless runoff volume is captured in the upstream portions of the watershed. The prolonged duration of storm flows leaving stormwater ponds can also increase erosion in streams [92]. Where stormwater pond effectiveness is limited, other GSI practices may be more attractive.

## 6. Filtration Practices

### 6.1. Overview

Stormwater sand filters can be used to remove suspended solids and solid-bound pollutants from stormwater runoff through physical straining and sedimentation. Sand filters were developed in the early 1980s by the City of Austin, Texas (USA) in response to climate and geographical constraints that limit or prevent the use of wetlands and other vegetation-based practices [165,166]. Infiltration practices were also not an option due to slow soil infiltration rates and concern for groundwater contamination. The first sand filters were installed above ground, but sand filters can also be installed underground where space is limited.

But whether above ground or underground, it is important to remember that captured contaminants do not simply disappear unless the filtration media (soil and sediments) is being periodically replaced with fresh, uncontaminated media as in the case of cartridge filtration systems. Otherwise, the pollutants are bound to accumulate in the media until the filter is overwhelmed and cannot capture any new contaminants. Conditions experienced by the filtration practice over time may allow certain captured contaminants to be released.

### 6.2. Intended Purpose

The advantage of surface sand filters instead of settling practices (including stormwater ponds) is not great, and surface sand filters are generally not specified due to the additional maintenance requirements that filtration brings with it. Underground sand filters are an appropriate application where land costs are high or land is not available. Pollutants carried by stormwater, whether particulate or dissolved, can be retained in soil structure via physical straining, cation exchange, adsorption to the surface of soil particles, microbial activity, and the formation of surface complexes. Iron filings, water treatment residuals, and other amendments, however, can be added to enhance the removal and retention of specific dissolved pollutants from runoff, such as phosphate [167–170]. Organic media such as compost, for example, is known to remove dissolved metals and petroleum hydrocarbons from runoff, although compost can also release phosphate. A denitrification filtration practice that is composed of a storage chamber underneath a biofiltration practice has also proven successful [171]. An enhanced filtration practice is one of the few manners in which dissolved pollutants [58,63] can be removed from runoff. Soil can also be used as a filtration media, however pollutant transport through the soil is ultimately affected by the organic content, microorganism activity in the vadose zone, porosity, infiltration capacity, moisture content, and other factors of the soil [172,173]. Thus, the transport and retention of pollutants through and in soil media is highly site-specific.

### 6.3. Potential Negative Consequences

Filtration is substantially more labor intensive than most other stormwater control measures, and this is one of the most frequent causes of negative consequences. Pre-treatment is normally required to keep gross solids, sand, and trash from clogging the filter surface, and public expectations of frequent trash removal may increase maintenance costs. In addition, inspection of the filter is desired after all major storms to determine if non-routine maintenance is required. A lack of maintenance can result in filter clogging and stormwater bypassing the filter, and any kind of maintenance is likely more difficult, dangerous, and expensive when the filtration practice is underground.

Release of previously-retained pollutants is possible when runoff characteristics, such as pH and ionic strength, change substantially. The media has a finite capacity to adsorb pollutants, and that capacity may become exhausted. In such cases, the pollutant(s) would not adsorb to the media but would continue to move with the water and through the filter. Metals may be removed by adsorption to soil media. Of the metals typically found in stormwater, lead and copper have the largest tendency to adsorb to solids, while zinc and cadmium are much less likely to adsorb [173]. Since metals are often bound to solid particles, removal of suspended solids can be an effective method of

reducing total metal concentrations in stormwater. Metals do not generally degrade to another element in the environment, however, and stormwater loading into filtration practices will result in metal accumulation. Using typical pollutant concentration in urban and roadway runoff and soil capacity estimates for bioretention practices, Davis et al. [174] estimated that, after 20 years, concentrations of cadmium, lead, and zinc would reach or exceed levels permitted by US EPA biosolids land application regulations. Once adsorbed to the surface of a soil particle, metals may not necessarily remain stationary and may be released from the soil surface if the water and soil conditions change to a lower pH and higher ionic strength.

### 6.4. Recommendations

Despite their increasing abundance, no specific literature was encountered evaluating the economic impact of filtration practices or their role in green gentrification. Further research on this topic is needed.

Filtration systems require regular inspection and maintenance, which typically includes raking the filter surface to break up fine particles that have collected and reduced the through-flow capacity and removal of the layer of fines that have collected on or near the filter surface and reduced the hydraulic capacity. More extensive maintenance will likely include replacing a portion of the sand bed and eventually the entire sand bed. Any inspection or maintenance of underground filtration practices will require appropriate safety procedures, and the need to enter underground practices can be reduced by designing them for maintenance using a vacuum truck on the surface.

Enhanced sand filters have additional inspection and maintenance issues regarding the enhancing agents. Because all enhancing agents have a finite capacity for removal, the effectiveness of an enhancing agent in a sand filter should periodically be determined. When the effectiveness has dropped to near or below satisfactory levels, enhancing agents should be removed and replaced. This may involve the removal and replacement of the entire sand media bed. Research is needed on the methodology of determining the effectiveness and the proper scheduling of enhanced sand filter replacement.

## 7. Infiltration Practices

### 7.1. Overview

Filtration practices that are not lined with an impervious layer can allow stormwater to percolate into the ground (infiltration). Infiltration is increasingly the strategy of choice for managing urban stormwater runoff, largely because infiltration reduces the volume of stormwater, and hence its pollutant loading to surface waters. Infiltration practices also include permeable pavements, which must be strategically located and diligently maintained to prevent permanent clogging. Several modeling experiments have shown that infiltration practices—when properly implemented—are capable of nearly restoring pre-development hydrology for smaller storms (approximately two to eight centimeters of rainfall depth), depending on the infiltration capacity of the soils [175,176]. This is a more permanent treatment of stormwater volume and stormwater contaminants than other GSI practices since stormwater runoff is converted into groundwater.

### 7.2. Intended Purpose

Several GSI practices incorporate infiltration as a primary stormwater treatment mechanism. Stormwater can first flow through permeable pavements or be routed directly to diverse infiltration practices. Infiltration chambers are sub-surface (underground) chambers that receive and store runoff, allowing it to infiltrate into the underlying soil. Infiltration trenches are trenches that are typically backfilled with large gravel. The void spaces in the gravel provide storage space for runoff that infiltrates into the existing soil. Infiltration basins are similar to detention ponds but, due to porous soil, are able to infiltrate larger volumes of runoff. A typical infiltration basin—also called a "retention pond" or "dry pond"— is covered with vegetation, which will also dry out the soil through evapotranspiration and has an outlet that is positioned at a level above the bottom of the basin. Unlined roadside drainage

ditches are another good example of infiltration practices, where water infiltrates into the soil as it runs into and along the drainage ditch.

*7.3. Potential Negative Consequences*

The success of an infiltration practice depends first and foremost on the infiltration rates of the underlying soils. New techniques are making it simpler and more affordable to take high-density infiltration rate measurements prior to construction [177–179]. If soils are not conducive to infiltration, a filtration practice may be more appropriate. Often times, infiltration is sought simply by redirecting, or "disconnecting," stormwater from impervious surfaces to more pervious surfaces like grass lawns. These downspout disconnections [180,181] function similarly to vegetated filter strips [182] in that they do not typically include engineered infiltration media characteristic of most infiltration practices and cannot infiltrate stormwater effectively when underlying soils have poor infiltration rates [183]. Construction and other activities can also greatly reduce infiltration rates through clogging or compaction [184]. But when infiltration does occur, other issues may arise around what contaminants are present in the infiltrated stormwater [185]. Of greatest concern are nitrate, toxic metals, pathogens, and chloride.

Nitrate is the most common nonpoint-source groundwater contaminant in the world [186]. In the United States, nitrate groundwater contamination occurs mainly in agricultural settings. Low nitrate concentrations in urban runoff generally present low to moderate risk of groundwater contamination, but, according to Pitt et al. [187], localized nitrate sources could increase this risk.

Soils typically retain metals within the top 30–50 centimeters, with some recommending at least 40 centimeters of unsaturated soil between the bottom of an infiltration practice and a groundwater source [188]. Metals have been detected in groundwater under infiltration practices when the groundwater is acidic [187], but the concentrations have typically been below water quality standards and, thus, are not currently considered as a threat. The risk of groundwater contamination by metals may change, however, as metal concentrations accumulate in the soil beneath infiltration practices over time.

All organic compounds have low or moderate contamination potential for subsurface infiltration with sedimentation as a pretreatment mechanism. With respect to potential groundwater contamination, it appears that most hydrocarbons are trapped in the first few centimeters of soil in infiltration basins [188,189]. The hydrocarbons may then be utilized by microorganisms in the soil [190]. This has led some to conclude that hydrocarbons pose little risk to groundwater contamination. In studies that have investigated the potential contamination of groundwater from infiltration of stormwater, however, it was not uncommon for groundwater to contain organic compounds, presumably from infiltrated stormwater [191–195]. In some cases, pollutant concentrations did exceed drinking water standards. Thus, considering the use of an infiltration practice should not occur without also considering the potential for groundwater contamination.

For residential and light commercial developments, pathogens (typically made up of bacteria and viruses) in stormwater are a primary pollutant of concern. Bacteria may be removed by straining at the soil surface and sorption to solid particles. Documented virus contamination of groundwater due to infiltration practices has occurred on Long Island, New York (USA) at sites where the distance between stormwater infiltration basins and the underground water table was less than ten meters [191]. For comparison, infiltration practices in Minnesota are only required to maintain a one-meter distance above the seasonal high water table or bedrock layer [196]. If removed from the water by soil, the ability of bacteria to survive is a function of factors such as temperature, pH, presence of metals, etc. Bacteria survive longer in acidic soils and in soils with large amounts of organic matter. Bacteria survival may be between two and three months, but survival for up to five years has been documented [191].

Because chloride is soluble, easily transported in surface and sub-surface flows, non-filterable, and does not readily adsorb to solids, it has a high potential for groundwater contamination [197]. Rather than being reduced, chloride concentrations typically increase as water moves through soil due to the leaching of salts into the water [191]. Novotny et al. [198] found that, in the Twin Cities

Metropolitan Area of Minnesota (USA), 72% of chloride from winter road salt application remained in the watershed, presumably being retained in soil and groundwater, with concentrations as high as 2000 mg/L found in shallow groundwater wells. High chloride concentrations can also cause the release of metals that are fixed to soil particles [199]. Once released, the metals are free to move with the water into the groundwater.

Placement of infiltration practices is critical because improper location of practices can lead to negative impacts on infrastructure. Because infiltration practices involve the concentration of stormwater into small areas, the hydraulic loading rate of soils is increased far above the loading rate under natural conditions. While the surface soil at the location of infiltration practices is designed to be able to handle the increased loading rate, the subsoils will generally be less suited for increased loading, and locally-elevated groundwater (mounding) could result beneath infiltration practices [200–202]. This elevated groundwater could then leak into underground infrastructure including sewers, water distribution lines, gas lines, electrical conduits, and basements of residential, commercial, and industrial buildings [203–207].

## 7.4. Recommendations

Due to the complex interactions between infiltration GSI and their surroundings, economic value associated with implementing GSI is not well understood and likely varies greatly with application setting and project goals. One study by Nordman et al. [208] suggests that infiltration practices may actually result in decreased property values because they do not typically offer the same aesthetic amenities as many other GSI practices. What role this may have in relation to gentrification is also not well understood and requires further research.

Although every infiltration practice must be considered on an individual basis with pollutant concentrations, runoff volumes, soil properties, and other factors taken into account, research indicates that the soil has the ability to capture many pollutants. Ebrahimian et al. [209] provide a detailed analysis of the factors that can affect infiltration rates in GSI and highlight steps that can be taken to improve them. Evidence suggests that hydrocarbons will be degraded by bacteria surrounding plant roots and metals will accumulate on the media, which will eventually need to be treated or excavated. In most cases, due to the large capacity for metals that most soils have, such action may not be needed for decades. Pathogens are also typically filtered by soil media, but this may not be true for viruses. Nitrate, although not retained by soils, is typically at concentrations in urban stormwater that raise little overall concern for nitrate contamination. The potential for groundwater contamination is usually greater for subsurface infiltration practices (such as infiltration chambers, infiltration trenches, etc.) as compared to surface infiltration practices (infiltration basins, etc.) due to inherent treatment mechanisms in the soil surface layer (microbial activity, organic material, etc.), but adopting regulations similar to those governing underground septic systems may address some of these concerns.

There is no practical way for an infiltration practice to remove chloride. Chloride is not retained in significant amounts by soil and is a long-term pollutant of concern for areas that use road salt during the winter to remove snow and ice from road surfaces. Applying chloride in the form of brine, as opposed to rock salt, can reduce chloride loads to streams by 45% [54]. Aside from modifying salting practice, it may be productive to prevent meltwater on salted surfaces from reaching GSI, at least during periods of winter melting.

For most pollutants discussed herein, cases of groundwater contamination due to stormwater infiltration have been documented. If care and consideration are given to the selection and placement of an infiltration practice, groundwater contamination can be minimized. Placement can also minimize impacts from elevated groundwater [206]. For example, a modeling study in Philadelphia (PA, USA) determined that infiltration practices need to be located at least three meters away from building foundations to reduce the potential for basement flooding and related issues [201]. In some cases, similar modeling of groundwater and pollutant transport may be warranted to verify that the potential risk of groundwater contamination is low. One challenge in proper site selection is being able to characterize subsurface properties without expensive and intrusive excavations (borings for instance). A possible

approach is to use non-intrusive methods such as ground-penetrating radar (GPR). Recent advances in the use of ground-penetrating radar have demonstrated that even thin, fine subsurface features, often the cause of unwanted groundwater mounding, can be identified with this methodology [210].

## 8. Rain Gardens and Green Roofs

### 8.1. Overview

Filtration or infiltration practices may take advantage of additional benefits by incorporating trees or vegetation in what are known as rain gardens. Rain gardens provide additional biological treatment that can permanently remove certain organic contaminants via biodegradation [190,211]. This depends, however, on the presence and health of the vegetation and soil microorganisms. Overuse of compost, fertilizer, irrigation, or other common landscaping practices, particularly when the planted species are not well-suited to the conditions of a rain garden, can result in accelerated nutrient accumulation in the rain garden media and lead to increased contaminant concentrations in stormwater flowing through the rain garden [212].

Rain gardens can also be placed on roofs, where they are called green roofs or living roofs. Most green roofs have shallower, more specialized media than rain gardens due to weight constraints and the need to support specialized species of vegetation, such as succulents, in extreme hydrologic conditions [213]. Contaminant accumulation in the media is less of an issue because green roofs are receiving rainfall rather than stormwater runoff, so incoming contaminant concentrations are relatively low. This also means that the relatively clean rainwater, however, has the potential to pick up organic matter and other contaminants from the green roof media and flow out with higher concentrations than the original rainwater [214,215]. Most rainwater should be absorbed by the media, except if the green roof is undersized, poorly maintained, or receiving very large rainfall events [216].

### 8.2. Intended Purpose

Various GSI practices incorporate vegetation as a defining characteristic and water treatment strategy. These include stormwater treatment wetlands, green roofs/living roofs, green walls/living walls [7], and rain gardens/bioretention cells. Rain gardens are shallow, vegetated depressions into which stormwater is directed for filtration—sometime also infiltration and groundwater recharge [217]—, while also allowing evapotranspiration. Green roofs do not allow for infiltration [218], but take advantage of limited land availability in urban settings. Their location on building rooftops enables green roofs to provide various additional benefits including heat island mitigation, air quality improvements, thermal and noise insulation, increased roof longevity, and many more [213,219].

Both rain gardens and green roofs have a combination of pollutant removal mechanisms [220], meaning that they do not fit into the infiltration (in the case of some rain gardens) or filtration practice categories despite typically having overlapping characteristics. There is a substantial body of work demonstrating the capacity of rain gardens to remove a wide range of pollutants from stormwater runoff including petroleum hydrocarbons [190,211], pesticides [221], toxic metals [28,222,223], nutrients [220,221], suspended solids [222], and bacteria [222,224]. And the potentially numerous benefits—in terms of pollutant removal, volume reduction, peak flow delay, and peak flow reduction—of rain gardens [220,222] and green roofs [219,225], combined with their aesthetic value, have resulted in their increased use for stormwater management and treatment.

### 8.3. Potential Negative Consequences

Rain gardens and green roofs are unique in that they are most commonly directly associated with individual properties or homes. Green roofs in particular are structurally connected to buildings and often implemented for aesthetic and social benefits [213] as well as energy efficiency [226] that lead to higher sale prices on the basis of perceived long-term savings [227–230], particularly if they have received environmental certification [208,231]. At the same time, the homes are less affordable

and may be less profitable for developers due to higher construction costs [232]. It is also worth noting that higher sale prices only result from green construction techniques for housing in the mid and high price ranges but not in the low price range [228].

Many rain gardens do not have lined bottoms and allow stormwater to freely infiltrate into the ground. This infiltration, however, poses potential risks to groundwater resources when pollutants are not completely captured by rain garden media. Pollutants can enter shallow groundwater beneath the rain garden and eventually reach deeper groundwater or even surface water bodies due to horizontal movement of groundwater, or "baseflow."

In addition, pollutants removed by physical and/or chemical processes (not biodegraded or taken up by plants), such as sorption and filtration, can potentially be rereleased due to shifts in water chemistry. For example, sorption is often a reversible process such that contaminants can detach from soil and other sorption sites when contaminant concentrations in the water decrease. Organic pollutant desorption can also be enhanced in rain gardens by chemicals released from plants [233]. Green roofs have also been found to release nutrients [214,215,234]. In rain gardens, Paus et al. [235] demonstrated that salt-laden runoff, from de-icing operations in cold climates, can mobilize previously-sorbed toxic metals from filtration media. Salt can also be harmful to rain garden vegetation and can increase the salinity of groundwaters receiving infiltrated stormwater runoff from streets and sidewalks in colder climates. The dissolved pollutants that are either not removed by the media or are remobilized after being removed initially may reach the water table and eventually contaminate drinking water wells and possibly even surface waters.

The overuse of compost in rain gardens and green roofs can cause phosphorus release [236,237]. Compost is often added to improve vegetation growth and toxic metal removal but contains and is capable of leaching phosphorus [28,238]. Hurley et al. [212] found that leaching rates of ammonium and phosphate increased with saturation time due to the development of anoxic conditions, while nitrate leaching rates decreased with saturation time due to denitrification under anoxic conditions.

To our knowledge, there is no direct evidence in the literature demonstrating the contamination of groundwater and associated water supplies has occurred due to infiltration from rain gardens [239,240]. The long-term fates and transport potentials of many pollutants in raingardens, however, are still poorly understood. The likelihood of pollutants reaching groundwater would be dependent on pollutant and subsurface characteristics. For example, strongly-sorbing pollutants (highly hydrophobic) are expected to be less likely to pass through bioretention media or to be remobilized.

The likelihood of particulate, or particle-associated, pollutants such as pathogenic bacteria, reaching the water table is dependent on their size compared to the infiltration media characteristics. As long as particles are larger than the pore space between adjacent media grains, they will be strained out of infiltrating stormwater and retained in the media. Therefore, larger particles and smaller media should result in increased particle capture, although there are limitations due to the potential for the media to become clogged and stop infiltrating altogether. Davis [241] found that only 43% and 47% of the suspended solids were removed (on average) from two bioretention cells monitored for twelve storm events. Additionally, rain gardens located in regions with Karst geology (characterized by increased drainage due to the presence of sinkholes and caves) would be expected to pose increased risk of contamination to groundwater supplies by a wide variety of stormwater-associated pollutants because subsurface fractures can accelerate infiltration and pollutant transport.

*8.4. Recommendations*

Green roofs are attractive for various reasons including urban heat island mitigation, air quality improvements, and thermal insulation. But when it comes to water quality benefits, it seems that many green roofs function most effectively as consciousness-raising tools [17] rather than stormwater treatment practices. Nevertheless, steps can be taken to make green roofs viable stormwater treatment practices as well. Nutrient-rich stormwater draining from green roofs can be routed to easily mown grass filter strips [136] by disconnecting downspouts from storm drains [180,181], harvesting rainwater

for reuse [242], or even implementing secondary GSI practices like rain gardens. Where these options are not viable, the effluent could be treated directly using various methods [214,234]. And when adequately sized, the issue of nutrient-rich effluent can be altogether avoided by fully capturing the rainfall volume and therefore treating 100% of the stormwater from most storms [216]. One emerging strategy to accomplish this is to couple green roofs with increased water storage capacity as so-called "green blue roofs" [219].

Design, construction, and maintenance greatly influence how rain gardens and green roofs perform. This is particularly true immediately following construction as vegetation will require special attention in order to become established. Selecting appropriate vegetation and properly sizing a system can greatly reduce the need for maintenance, improve overall performance, and increase longevity. Identifying specific treatment targets can also make success more likely. For example, organic matter, often in the form of compost, is effective at removing toxic metals and various other contaminants [28]. But the potential to release nutrients means that compost would be counterproductive in situations where the treatment priority is nutrients. In those cases, compost should not be applied unless absolutely necessary for vegetation growth, in which case it could be applied sparingly and directly to the root zone of each plant rather than as a continuous layer. Low-phosphorus compost can also be sourced [212].

## 9. Discussion

The hydrologic impact of any GSI practice can be evaluated by considering the entire flow path of water through a watershed (Figure 3). Following our decision-making framework (Figure 2) can simplify the consideration of diverse factors that interact with GSI in complex ways and ultimately influence the potential for unintended consequences to occur. This discussion will progress through that framework as an example of how it could be applied to any GSI implementation effort in order to select the best GSI practice type for the job, maximize its potential benefits, and minimize its potential consequences.

### 9.1. Identify Goals

Will the practice treat water quality but not water volume or peak flow rate, or only peak flow rate, or all three? Will the practice add organic matter and other contaminants to relatively clean rainwater or stormwater? It is likewise vital to consider what lies downstream of each GSI practice. If the purpose of installing GSI is to protect a lake from accumulating phosphorus and experiencing algal blooms, for example, planting trees along streets and installing green roofs may actually lead to more nutrients flowing into the lake [62,214]. Infiltration practices could also reduce the ability of phosphorus to be flushed out of the lake during large rain events [236]. It is therefore important to have as complete an understanding as possible of the factors at play within a project in order to increase the likelihood of meeting project goals.

### 9.2. Prioritize Goals

Water quality goals exist in a broader context of ecosystems and communities, and failing to consider the broader impacts of GSI projects can prevent the overarching goal of providing clean water to all. Particularly within the framework of urban greening or urban redevelopment (also called "urban revitalization" or "urban renewal"), GSI implementers must consider how GSI practices affect communities, especially residents that may have been excluded from the benefits of past investments in GSI or be skeptical of interventions that are not perceived as "of" or "for" their communities [110,243]. More research is needed to understand who stands to benefit from GSI, especially when it is embedded in larger developments [33,34]. Integrating a socioeconomic lens from the beginning of GSI project assessment can also help address the often-disconnected nature of overlapping and conflicting goals and strategies of various groups—too often prioritized on the basis of economics alone [37]—within a single planning effort.

### 9.3. Characterize Loading

It is also important to consider what contaminants each GSI practice is expected to treat [66,67]. Whether the contaminants are the targets for treatment or not, they will interact with GSI practices in some way and either be treated, accumulate, or pass through the practice. Further research is needed to understand which contaminants are effectively treated by GSI and which may impair its overall functionality (for example, chloride). Considering the ultimate fate of each contaminant is an important step toward reducing potential unintended consequences.

Certain conditions may make it infeasible to fully treat the stormwater load entering a project site. This could be due to limitations from project funding, available area for the GSI practice, stakeholder preferences, or other unique constraints. A growing body of research on undersized GSI practices suggests that they can still provide effective treatment to at least part of the stormwater load [180,244], although additional maintenance and operation requirements will need to be taken into consideration. Undersizing may be less suitable for harder-to-maintain GSI practices, such as wetlands [245] and ponds [246], and practices where treatment is greatly limited by reduced temporary storage volume, such as rain gardens [247] and green roofs [216]. As long as care is taken so that no inadvertent harm is done, some treatment is still better than no treatment.

### 9.4. Identify Strategies

The literature discusses which practices have better removal rates for which contaminants, but it is also crucial to consider the ultimate fate of these contaminants. Some small, contaminant-rich outflows could be routed into secondary GSI practices to remove the contaminants or treat the remaining water volumes. Will contaminants accumulate in pond sediments and make maintenance prohibitively expensive? Will they accumulate in filtration/infiltration/rain garden/green roof media and require that the media eventually be replaced (often at a cost greater than the initial GSI practice construction cost [72])? The benefits of short-sighted "band-aid" solutions can be especially short-lived or even detrimental in the long run. But every tool must fit the task at hand, and no solution can be expected to solve everything.

### 9.5. Analyze Strategies

Proper GSI design and maintenance must prioritize public welfare, health, and safety [248]. This includes the obvious considerations for preventing personal injury and destruction of property but also includes the anticipation of issues that could arise. For example, in roadside GSI applications, overgrown vegetation can obstruct vehicular lines of sight or offer hiding places to potential assailants on dimly lit streets. Pruning, thinning, and limiting the height of vegetation can easily overcome these negative consequences, but only when proper training and funding are provided for regular maintenance. Similarly, GSI practices must be properly maintained to prevent mosquito breeding habitats or invasive species establishment. There are also less obvious considerations, such as the risk of green spaces increasing pollen and other allergens in urban settings and the possibility of stormwater from rain gardens entering adjacent basements, leaking sewer mains, or other infrastructure. These and other impacts may intensify as GSI practices age.

Malfunctioning GSI practices also have the potential to act as sources of contaminants rather than sinks. These malfunctions can largely be avoided, however, with proper design, construction, and maintenance. While construction risks can never be completely removed, they can be greatly reduced when practitioners have proper training and experience [249].

### 9.6. Re-Evaluate

Few research studies have included full lifecycle analyses, and so the long-term maintenance needs of GSI practices are poorly understood, poorly defined, and poorly funded. Continuous monitoring is becoming increasingly affordable and accessible with the advent of low-cost, DIY systems supported

by dedicated communities (for example, EnviroDIY [250] and Open-Storm [251]). It is now possible to supplement traditional inspections with remote sensors collecting real-time data on key performance indicators as well as inexpensive video recording [252] and even drones [253]. Traditionally, most monitoring and sampling is done on sunny days during the warm season, but GSI performance varies seasonally as does pollutant loading (for example, nutrients trapped in snow and released during snowmelt while plants are still dormant and the soil may be frozen [136]). In addition to enabling high-frequency and continuous monitoring, low-cost, DIY (do-it-yourself) systems introduce the opportunity for adaptive or even active management to improve GSI functionality but also introduce a new level of concerns ranging from electrical failures to cyber security [254]. And yet, these threats are not entirely new. Legacy supervisory control and data acquisition (SCADA) systems that are not connected to the internet in water supply and drainage systems lend a false sense of security from cyber-attacks but remain vulnerable at other points (such as the use of unsecured radio channels) [252]. It is always necessary to constantly re-evaluate decisions and adapt strategies to changing conditions as time passes and scientific understanding improves [255], and continuous monitoring can facilitate this adaptive management and resolve unintended consequences as they emerge.

Infrastructure improvements are necessary and could follow a number of different paths. Infrastructure will be tested and challenged over its lifespan; it will require monitoring and maintenance, and it may fail. The goal is to ensure that failure, however unlikely and whether in anticipated ways or unanticipated ways, has minimal negative consequences by erring on the side of caution and incorporating designed redundancies, resilience, and factors of safety.

## 10. Conclusions

GSI has the potential to generate multiple water quality, social, economic, and ecosystem benefits [7,25,44,256]. Urban trees can treat stormwater through evapotranspiration [104] and also provide species and recreational habitat [44] while cooling urban areas in subtropical climates [105,119]. Stormwater ponds provide temporary flood storage [124], pollutant removal on the basis of particle settling [136] and denitrification [75,137–139], and biodiversity and recreational opportunities [14,140]. Filtration practices are frequent components of GSI where surface land area is limited and can remove specific dissolved pollutants from runoff, such as phosphate [58,63,167–170]. Infiltration is another increasingly common component of GSI and has the capacity to "permanently" remove large volumes of stormwater and associated pollutants according to the infiltration capacity of soils [175,176]. Rain Gardens bring together the benefits of urban vegetation, filtration, and potentially infiltration to realize many diverse benefits [220,222]. Green roofs have similar qualities [225] and offer additional, unique benefits related to being located on top of structures, such as thermal and noise insulation [219]. The actual functions of GSI (whether physical, chemical, biological, or social) are more significant than the mere perception of having installed GSI, and therefore steps must be taken to ensure that the actual functions are the desired functions. Not only that, but all of these functions are fundamentally interconnected and must be considered simultaneously in order to effectively meet goals [27].

But, all of these benefits depend on regular maintenance, which must be included in plans, budgets, and regular operations to prevent GSI practice failure. If unintended consequences begin to outweigh potential benefits, some form of revitalization is required. And in cases where revitalization is impractical or impossible, decommissioning of the practice altogether may become necessary.

The best tool to solve a problem in one setting may not be as effective in another. There are good reasons to implement green roofs [219], but they may not be the preferred strategy where nutrient reduction is the priority [215]. Similarly, there are good reasons to plant trees in urban environments [257], but doing so may not be an effective urban heat island mitigation strategy in tropical climates [121]. Understanding the strengths and weaknesses of each GSI practice is crucial to effectively managing a watershed.

The true potential for unintended consequences to occur is unknown, particularly with the newest GSI practice types, but much of the fundamental understanding already exists within other fields of

study. As our synthesis reveals, GSI interacts with a complex network of urban hydrology (Figure 3) and can have many impacts on water resources, other infrastructure, and the people who use both. The next step in sustainable planning would be to go beyond the watershed and consider the role of GSI as part of an ecosystem [258]. By approaching each new GSI project from a logical decision-making framework (Figure 2), it is possible to consider the various factors that determine GSI performance and impacts as well as the various groups of people who will benefit from GSI. With proper planning, many potential negative impacts can be minimized or altogether avoided.

The necessity and advantages of investment in GSI are apparent, and so too are the needs for monitoring of long-term performance and funding of proper maintenance for GSI. In the United States, however, research funding to do so is severely limited at the federal level and varies greatly from state to state. Without supporting fundamental research, there is a continued risk of wasting money in ineffective ways. We echo Prudencio and Null [14] in calling for further research to quantify ecosystem services, and we emphasize the additional need to quantify ecosystem disservices. Only then can we truly realize green infrastructure that maximizes infrastructure, ecosystem, and community services [7].

**Author Contributions:** Conceptualization, V.J.T. and J.S.G.; writing—original draft preparation, V.J.T., P.T.W., J.S.G., R.M.H., L.A.B., M.R.K., and J.C.F.; writing—review and editing, V.J.T., M.R.K., B.L.K., J.C.F., J.S.G., R.M.H., L.A.B., P.T.W., and J.L.N.; visualization, V.J.T. and L.A.B.; project administration, V.J.T.; supervision, J.S.G. and J.C.F. All authors have read and agreed to the published version of the manuscript.

**Funding:** This research received no external funding. The first author was supported by a fellowship from the National Science Foundation (grant number 00039202). The ninth author was partially supported by the US Department of Agriculture National Institute of Food and Agriculture (Hatch/Multistate project MN 12-109).

**Acknowledgments:** The authors wish to acknowledge the contributions and support of various individuals and groups without which this study would not have been possible. Firstly, we respectfully acknowledge that the lands on which this study occurred are the original homelands of the Dakota and Ojibwe Nations. And we aspire to honor and respect the Indigenous peoples who were forcibly removed from and are still connected to this territory by owning our part in their continued displacement. The authors also wish to thank the following: Katie Wilson, Biosciences Liaison Librarian and Scientific Data Curator at the University of Minnesota Libraries for assisting with the term usage analysis; Sarah "Winnie" Winikoff, Graduate Student in the Department of Ecology, Evolution, and Behavior at the University of Minnesota, for helping to conceptualize the wind sheltering reduction scheme for stormwater ponds; M.B., Writing Program Coordinator at the University of Minnesota Graduate School Diversity Office for assisting with paper organization; and Micaela Magee, Project Coordinator at Datatrend Technologies, for assisting with manuscript clarity and enhancing visualization design. Lastly, this manuscript would not have been possible without the ideas incorporated from personal conversations and research presentations by the following: Cameron Twombly, former Graduate Student at the Department of Civil and Environmental Engineering of the University of Vermont; A.E., Researcher at the St. Anthony Falls Laboratory of the University of Minnesota; B.J., Researcher at the St. Anthony Falls Laboratory and Department of Ecology, Evolution, and Behavior of the University of Minnesota; A.C., Assistant Professor at the Department of Entomology and Wildlife Ecology of the University of Delaware; J.H., Research Assistant Professor and Program Director at the Stormwater Center of the University of New Hampshire; E.F.-B., Associate Professor at the Department of Civil, Environmental, and Ocean Engineering of the Stevens Institute of Technology; S.S., Principal at Geosyntec Consultants; and Ross Bintner, Engineering Services Manager at the City of Edina, Minnesota; Lauren Williams, Landscape Designer at DesignJones LLC; Natalie Carmen, Project Engineer at Stewart; Shahram Missaghi, Water Resources Regulatory Coordinator at Minneapolis Public Works; and M.J., Senior Associate at Hazen and Sawyer.

**Conflicts of Interest:** The authors declare no conflict of interest.

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
