# Peer review of "It Is Not Easy Being Green: Recognizing Unintended Consequences of Green Stormwater Infrastructure"

_water, doi:10.3390/w12020522_

Round 1

Reviewer 1 Report

Although well written, this manuscript suffers from a lack of focus and/or clarity of purpose. It is unclear if the intention is a review of maintenance challenges, advocacy for a new SGI management framework or an expose of unintended consequences (or all of the above). There is certainly need for better long-term management of green infrastructure, and the proposed framework has merit – however the ‘framing’ of the manuscript means these issues/concepts are not articulated clearly. I offer only a few broader comments below, given the recommendation for major revision.

This paper seems to suffer from an identify crisis. It initially identifies as an examination of the ‘unintended consequences’ of constructing green infrastructure, however significant sections of the manuscript (especially the discussion) relate primarily to the decision-making framework proposed by the authors. The two elements do not seem well integrated, or at least, it’s not clear to this reader how they assimilate. Furthermore, the introduction introduces the concepts of ‘marginalized communities’ and ‘environmental justice’, but there is little mention of these during the analysis of SGI types or in the discussion. Rather, the consequences they discuss are largely technical. I don’t consider the manuscript to be a true examination of the “unintended consequences” of SGI. Many of the “Potential Negative Consequences” described by the authors read more like management and maintenance issues, or at the least, design considerations. Many of these I consider to be well known by practitioners and well referenced in the scientific literature. As a simple example, I might have expected that the unintended negative consequence of treatment ponds could be an increase in mosquito populations. The accumulation of sediments and heavy metals is expected (they are often designed to do this) and their leaching a consequence of poor or under management. While the paper does provide a reasonable discussion of some the negative consequences of SGI, it fails to assess the level of impact these consequences have. That is, it does not weigh them against the benefits provided by SGI. The sections of the manuscript that explores this issue might be better written as a cost-benefit style analysis – assessing if the potential negatives out way the benefits. Similarly, some of the negative consequences are recognized limitations of SGIs – that they are nor a ‘cure all’. But that does not mean they have no value. For example, the authors are correct in stating that raingardens might not remove all pollutants entering them, but they must also acknowledge that without any treatment ALL the pollutants would otherwise enter the receiving waterway.

Other comments

The authors state that their findings apply to temperate regions of wealthier countries. But in some of their examples of negative consequences, they cite examples from tropical countries. Also, the paper seems very USA centric in the use the design and use SGI. I would recommend that the introductory sections outlining the issues around stormwater and its management be reduced substantially. These issues are generally well understood in the industry and there is now a significant body of scientific literature. The paper might read better if the authors were to make broader, encompassing statements that are supported by citations. Many statements or assertions in the manuscript are made without enough supporting citations. For example, lines 129-133.

Reviewer 3 Report

More detailed comments to authors are below:

Introduction:

Please provide more background studies on the limited/ negative effect of GI on stream water quantity and quality.

Line 86: “…, and just” an incomplete sentence.

Line 98: “most widely implemented GSI practices” in the US or worldwide?

Line 100: What were the criteria for authors to define “wealthier countries”?

Line 105: “each GSI practices fits into the hydrology” What about water quality?

Figure 2: Characterize loading circle: intensity and frequency of precipitation is a driving force and it is crucial to characterize contaminants load. I would suggest adding this.

Overall: The objectives/goals are not clearly stated in the introduction part.

Urban stormwater:

Line 112: “ the watershed”, urban or suburban watershed?

Line 125: pets can be added as one of the sources of nutrients load in urban watersheds.

Line 142: It is really confusing to entitle the section 3 as “Green Infrastructure”. Because all the other practices including green roofs, rain gardens, permeable pavement can define under this category. I would recommend changing the title to a more clear one.

Line 319: please provide appropriate citations.

Overall: All recommendations sections mainly expanded from the “intended purpose” and “potential negative consequences” without providing scientific recommendations based on GSI manuals and literature.

Discussion:

Line 711: “will the practice treat water quality…” or both?

Round 2

Reviewer 2 Report

I'm satisfied with the authors’ corrections. They reply to all my comments

Reviewer 3 Report

-